# Birth Outcomes in DES Children and Grandchildren: A Multigenerational National Cohort Study on Informative Families

**DOI:** 10.3390/ijerph20032542

**Published:** 2023-01-31

**Authors:** Laura Gaspari, Marie-Odile Soyer-Gobillard, Nathalie Rincheval, Françoise Paris, Nicolas Kalfa, Samir Hamamah, Charles Sultan

**Affiliations:** 1CHU Montpellier, Université de Montpellier, Unité d’Endocrinologie-Gynécologie Pédiatrique, Service de Pédiatrie, 34295 Montpellier, France; 2CHU Montpellier, Université de Montpellier, Centre de Référence Maladies Rares du Développement Génital, Constitutif Sud, Hôpital Lapeyronie, 34295 Montpellier, France; 3Université de Montpellier, INSERM 1203, DEFE, 34295 Montpellier, France; 4CNRS, Sorbonne Université, 75005 Paris, France; 5Association Hhorages-France, 66100 Perpignan, France; 6Equipe de Recherche AESIO SANTE, Clinique Beausoleil, 34070 Montpellier, France; 7CHU Montpellier, Université de Montpellier, Département de Chirurgie Pédiatrique, Hôpital Lapeyronie, 34295 Montpellier, France; 8CHU Montpellier, Université de Montpellier, Service de Biologie de la Reproduction, 34295 Montpellier, France

**Keywords:** diethylstilbestrol (DES), multigenerational transmission, prenatal exposure, endocrine-disrupting chemicals (EDC), birthweight, epigenetic

## Abstract

Objective: Diethylstilbestrol (DES), a potent synthetic nonsteroidal estrogen belonging to the family of endocrine disrupting chemicals (EDCs), can cross the placenta and may cause permanent adverse health effects in the exposed mothers, their children (exposed in utero), and also their grandchildren through germline contribution to the zygote. This study evaluated pregnancy duration and birthweight (BW) variations in the children and grandchildren born before, during, and after maternal DES treatment in the same informative families, to rule out genetic, endocrine, and environmental factors. Design and setting: Nationwide retrospective observational study on 529 families of DES-treated women registered at the HHORAGES-France Association. The inclusion criteria were: (i) women with at least three pregnancies and three viable children among whom the first was not exposed in utero to DES, followed by one or more children with fetal exposure to DES, and then by one or more children born after DES treatment; (ii) women with at least one pre-DES or post-DES grandchild and one DES grandchild; (iii) confirmed data on total DES dose. Women with severe pathologies or whose illness status, habitat, lifestyle habits, profession, treatment changed between pregnancies, and all mothers who reported pregnancy-related problems, were excluded. Results: In all, 74 women met all criteria. The preterm birth (PTB) rate was 2.7% in pre-DES, 14.9% in DES, and 10.8% in post-DES children (Cochran-Armitage test for trend, *p* = 0.0095). The mean BW was higher in DES than pre-DES full-term neonates (≥37 weeks of gestation) (*p* = 0.007). In grandchildren, BW was not different, whereas the PTB and low BW rates were slightly increased in children of DES women. Conclusions: These data within the same informative families show the DES impact on BW and PTB in DES and post-DES children and grandchildren. In particular, mean BW was higher in DES than pre-DES full-term neonates. This result may be in opposition to previous data from American cohorts, which reported lower BW in DES children, but is consistent with animal study. Our retrospective observational study highlights a multigenerational and likely transgenerational effect of this EDC in humans.

## 1. Introduction

Diethylstilbestrol (DES), a potent synthetic nonsteroidal estrogen that belongs to the family of endocrine disrupting chemicals (EDCs) was widely prescribed to pregnant women from the late 1930s to the 1970s [1]. Although the placenta is considered an important barrier for fetal protection during pregnancy [1], DES, like other EDCs, can cross the placenta and may cause permanent adverse health effects in the exposed mothers (DES-treated mothers), their children who were directly exposed in utero (DES children), and also their grandchildren through the male or female germline contribution to the zygote (DES grandchildren) [2]. These multigenerational adverse effects may be explained by the observation that in near-term pregnant rats, DES concentration levels after 30 min of constant intravenous carbon-14 (^14^C) DES infusion were 2–3 and 20–25 times higher in fetal plasma and fetal reproductive tract than in maternal plasma [3].

Although the real number of women exposed to DES is still unknown, several million individuals have been prenatally exposed worldwide, including approximately 200,000 children in France [4]. It is well-known that in utero exposure to DES induces a wide range of reproductive tract abnormalities in ‘DES daughters’, such as alterations of Müllerian duct development, fertility problems, ectopic pregnancies, miscarriages, premature births, endometriosis and cancers, particularly clear cell adenocarcinoma of the vagina and cervix in girls and young women [5,6,7,8,9,10,11,12,13,14,15]. In ‘DES sons’, epididymal cysts, hypospadias, cryptorchidism, hypoplastic testis and micro-penis have been reported [16,17]. Besides such somatic effects, prenatal DES exposure has been associated also with psychiatric disorders, such as severe depression, behavioral disorders, eating disorders, schizophrenia, bipolar disorders, anxiety, suicide, and suicide attempts, as well as neurodevelopmental disorders, including autism spectrum disorder [18,19,20,21]. Conversely, only a few studies have reported DES effects on preterm birth (PTB) and birthweight (BW) [7,22,23,24] in DES children and the following generations. Yet, according to the Developmental Origins of Health and Disease hypothesis, alterations in fetal growth (i.e., low birth weight, LBW) play an important role in the risk of developing diseases during adulthood (e.g., metabolic disorders, hypertension, stroke, coronary heart disease and related disorders) [25,26,27]. Moreover, all studies on fetal DES effects compared PTB and BW in in utero exposed children and in the general population, but did not take into account potential confounding variables (e.g., genetic background, habitat, and job).

Therefore, the aim of this retrospective observational study was to determine pregnancy duration and BW variations in all children (i.e., exposed in utero and also those born before and after the DES pregnancy) of DES-treated mothers in the same informative families, thus potentially ruling out genetic, endocrine, and environmental factors that could have been associated with BW variations. This analysis concerned also the DES-treated women’s grandchildren (i.e., one of their parents was exposed in utero, or was born before/after the DES pregnancy) to highlight a possible multigenerational DES effect on pregnancy duration and BW.

## 2. Results

Among the 529 women and their family registered in the HHORAGES-France Association database, 57 declined to participate. After exclusion of women with incomplete questionnaires (n = 58) or without documented DES exposure (n = 68), 136 women met all three inclusion criteria, and 74 of them were included, because they did not have any of the study exclusion criteria (i.e., severe pathologies, changes in illness status, habitat, lifestyle habits/job, or treatments between pregnancies, or pregnancy-related problems) (Table 1). The flowchart showing the included and excluded patients (with the reasons) is in Figure 1.

The 74 DES-exposed mothers (G0) were born between 1939 and 1945. They were prescribed DES to prevent miscarriage or PTB and also for comfort. DES was never prescribed for infertility in our sample. The treatment was initiated in the first (n = 60 women; 81.08%), second (n = 13 women; 17.57%), or third (n = 1 woman; 1.35%) trimester of pregnancy. When mothers had more than one DES pregnancy, the schedule and prescribed DES doses were the same for all pregnancies. The mean prescribed DES dose was 3982.4 ± 412.4 mg. The main characteristics of G0 mothers and their children are listed in Table 2 and Table 3. The mean interpregnancy interval (IPI) was 1.4 ± 0.4 years between the birth date of the pre-DES pregnancy and the estimated conception of the DES pregnancy and 1.4 ± 0.5 years between the DES and post-DES pregnancies. Specifically, the mean IPI was 1.4 ± 0.4 years between the pre-DES and DES pregnancy, 1.4 ± 0.4 years between the first DES and the second DES pregnancy, when present (n = 20), 1.4 ± 0.5 years between the DES and first post-DES pregnancy, and 1.4 ± 0.4 years between the first and the second post-DES pregnancy (n = 19). The G1 sample included 74 (37 boys and 37 girls) pre-DES, 94 (46 boys and 48 girls) DES, and 93 (43 boys and 50 girls) post-DES children, respectively (Table 2). G1 children were born between 1960 and 1972 (pre-DES children), between 1962 and 1975 (DES children), and between 1965 and 1978 (post-DES children). The PTB (<37 gestational weeks, GW) rate was 2.70% in pre-DES children, and increased to 14.9% and 10.8% in DES and post-DES children, respectively (Table 3). The LBW rate was 0% in pre-DES children and increased to 2.9% and 2.5% in DES and post-DES children, respectively. The high birth weight (HBW) rate was 4.2% in pre-DES children and increased to 11.6% and 22.5% in DES and post-DES children, respectively.

The mean BW for full-term newborns (i.e., ≥37 GW) was higher in DES than pre-DES children (*p* = 0.0071), but was similar in DES and post-DES children (*p* = 0.3645). The Cochran-Armitage test found a positive trend in the DES and post-DES groups for PTB (*p* = 0.0095), LBW (*p* = 0.0006) and HBW (*p* = 0.0033) (Table 3). In addition, multivariate mixed modeling to select determinants of the three defined outcomes (PTB, LBW, and HBW) showed that none of the variables included in our model had a significant effect on PTB (DES exposure was not significant probably because of the small sample size) (Table 4). Conversely, LBW was associated with GW (*p* < 0.0001) and DES exposure (*p* = 0.0269). HBW was associated with DES exposure (*p* = 0.0005) and number of pregnancies (*p* = 0.0087). The DES doses prescribed to the G0 mothers were not associated with PTB, LBW and HBW.

The characteristics of the G1 mothers (G1 daughters or partners of G1 sons, when adults) and the G2 sample (grandchildren) are shown in Figure 1 and Table 5. The G2 sample was divided in two groups in function of the G1 parent (father or mother) (pre-DES, DES, or post-DES): “paternal germline” and “maternal germline”. Among the 261 G1 children (pre-DES, DES, and post-DES), 23 never had children (8.8%), 8 partners declined to participate (3.1%), and 65 mothers were excluded for severe comorbidities and/or pregnancy-related problems (24.9%) with known effects on BW (Table 5). None of the G1 mothers reported changes in illness status, habitat, lifestyle, habits/job at risk of EDC exposure, treatments (i.e., antiepileptic drugs) between pregnancies or were prescribed DES during their pregnancies. Smoking was reported by two mothers in the G1 pre-DES group (5.0% of 40), four in the G1 DES group (4.9% of 82), and three in the G1 post-DES group (7.0% of 43).

The G2 (grandchildren) group included 49 (28 boys and 21 girls) pre-DES, 96 (44 boys and 52 girls) DES, and 47 (21 boys and 26 girls) post-DES children, respectively (Table 6). G2 grandchildren were born between 1979 and 2002 (pre-DES), between 1981 and 2006 (DES), and between 1984 and 2007 (post-DES).

Overall, the IPI was 1.9 ± 0.3 years in the paternal germline subgroup, when a second child was present (n = 4), and 1.5 ± 0.5 years in the maternal germline subgroup, when a second child was present (n = 5). In the DES group, the IPI was 1.5 ± 0.5 years in the paternal germline subgroup, when a second child was present (n = 7), and 2.2 ± 0.4 years in the maternal germline subgroup, when a second child was present (n = 7). In the post-DES group, the IPI was 2.0 ± 0.3 years in the paternal germline subgroup, when a second child was present (n = 2), and 1.90 years in the maternal germline subgroup, when a second child was present (n = 1).

In the three paternal germline subgroups, the PTB rates were 8.7% (pre-DES), 0% (DES), and 10% (post-DES). In the three maternal germline subgroups, the PTB rates were 7.7% (pre-DES), 14.8% (DES), and 14.8% (post-DES).

BW, BW in full-term newborns, LBW and HBW rates of G2 grandchildren were comparable in all groups.

## 3. Discussion

This retrospective observational study suggests that prenatal DES exposure is associated with a higher risk of PTB and with a significant increase in BW and HBW rate in full-term newborns. This analysis was carried out in the same informative families, thus potentially ruling out genetic, endocrine, and environmental factors that could be associated with the pregnancy outcomes under study. We obtained similar results also for post-DES children. In grandchildren, PTB, LBW, and HBW rates were slightly increased in children born from DES and post-DES daughters.

The PTB rate was 2.7% in pre-DES children and increased to 14.9% in DES children. Although we could not demonstrate a direct effect of DES exposure on PTB, probably because of the small sample size, these results further support a fetal effect of DES on PTB. Indeed, a meta-analysis in the general population reported higher odds of PTB among nulliparous mothers compared with women in their second pregnancy (OR 1.15, 95% CI [1.13, 1.16]) [28]. PTB rate was higher also in post-DES children (10.8%), suggesting post-exposure effects. Indeed, DES is mainly metabolized to its catechol quinone that reacts with DNA to form adducts that are stored in the adipose tissue [29]. This could suggest a risk of exposure even in the absence of a direct treatment. Actually, research on DES exposure during fetal life and PTB is still limited. Dieckmann et al. compared 800 pregnant women who took graduated amounts of DES according to the schedule suggested by Smith et al. [30] and 800 control pregnant women (the Chicago cohort). They found that the mean pregnancy length was shorter in DES-exposed primiparas and multiparas compared with their controls (38.7 GW vs. 39.3 GW, and 38.6 GW vs. 39.4 GW; *p* < 0.01) [22]. Moreover, they reported that the overall PTB rate was 6.43% in the DES group and 3.97% in controls (Risk Ratio (RR) 1.62, 95% CI [1.06–2.48]): 5.1% vs. 3.6% in primiparas and 8.4% vs. 4.9% in multiparas [22,23]. Similarly, Ferguson compared 200 patients who took incremental doses of DES according to the regimen described by Smith et al. [30] and 200 controls, and found that the mean pregnancy duration was longer in controls than in the DES group (38.1 vs. 36.6 GW) [24]. Conversely, the overall PTB rate was 25.79% in the DES group and 30.51% in controls (RR 0.84, 95% CI [0.61–1.16]) [23,24]. More recently, Hatch et al. reported that PTB incidence was 12.8% (OR, 2.54, 95% CI 1.90–3.40) and 14.6% (OR, 3.29, 95% CI 2.45–4.33) in American low- and high-dose DES cohorts, respectively, compared with 5.0% in control women [7].

In our study, BW was significantly higher in DES than pre-DES full-term newborns (*p* = 0.0071), but was similar in DES and post-DES babies (*p* = 0.3645). Dieckmann et al. found no BW differences in the Chicago Cohort [22], while Ferguson reported that the mean BW was 3080 g in the control group and 2997 g in the DES group (not statistically significant) [24]. Moreover, Hatch et al. reported that after adjusting for cohort and gestational age, the mean BW difference between DES and non-DES children, was −105 g (95% CI −134, −76) [7].

Moreover, both LBW and HBW rates increased in DES and post-DES children compared with pre-DES children (Table 3). Only two studies analyzed LBW rate in DES children and reported LBW rates of 7.39% and 18.42% in DES children and of 4.90% and 12.81% in controls, respectively, but the risk ratio was not significant [23,24]. Moreover, Dieckmann et al. reported HBW rates of 0.7% and 1.9% in DES-exposed primiparas and multiparas and of 0.5% and 1.4% in control primiparas and multiparas [15].

The possible mechanisms underlying the higher risk of PTB and of LBW reported in most studies on DES children remain speculative. According to Hatch et al., DES may interfere with the delicate balance of pregnancy hormones, such as the ratio of estradiol to estriol (two estrogens), and this might alter the timing of parturition [7,31]. DES may also induce maternal and fetal stress, leading to an increase in the secretion of corticotropin-releasing hormone, associated with preterm labor and the premature rupture of membranes [7,32,33,34,35]. In addition, the higher BW found in our cohort of DES children compared with previous literature data was predictable. Indeed, the outcomes observed in women treated with DES during pregnancy can be accurately replicated in the mouse [36]. For instance, in pregnant mice exposed to a 4-log range of DES concentrations from day 11 to day 17 of gestation, Palanza et al. did not observe any PTB, but delayed parturition in 50% and 100% of mice treated with 10 µg and 100 µg DES, respectively [37]. DES did not have any significant effect on BW; however, the dose-response relationship was complex (non-monotonic biphasic), and opposite effects (i.e., higher, or lower BW) were observed as DES doses increased [37]. In women, the total DES doses prescribed ranged between <100 mg and up to 46,600 mg [38]. In the USA, “high dose cohorts” with median DES doses of 12,442 (Chicago), 8575 (Boston), and 7550 mg (California) have been monitored, as well as “low dose cohorts” with median doses of 3175 (Wisconsin), 2572 (Texas), and 1520 mg (Minnesota) [39,40]. The median total dose in our HHORAGES-France cohort and in another French cohort described by Tournaire et al. was ~4000 mg [40]. Therefore, in our cohort, fetuses were exposed to DES doses between those of the American “low-dose” and “high-dose cohorts”. We may hypothesize that BW variations may follow a similar non-monotonic biphasic relationship as observed in the mouse model. This might explain why in our cohort, BW in full-term newborns was higher in the DES than pre-DES group, unlike previous findings in American cohorts [7,24].

In our G2 sample, PTB rates were 14.8% in children of G1 daughters exposed in utero to DES, 7.7% in children of pre-DES G1 daughters, and 14.8% in post-DES G1 daughters. The same PTB rates in children of DES and post-DES G1 women may suggest the hypothesis of a chronic DES contamination in the G0 mothers. Recently, few studies associated DES exposure during pregnancy to adverse health outcomes in generations beyond the one directly exposed in utero [41]. It has been reported that DES use in pregnancy is associated with higher PTB risk in the second generation [7,10]. Hoover et al. combined data from three studies initiated in the 1970s (the DESAD Cohort, the Dieckmann cohort, and the Women’s Health Study cohort) with long-term follow-up data from 4653 DES-exposed women and 1927 unexposed age-matched controls, and found PTB rates of 26.16% for DES women and 8.08% for controls (HR: 4.68; 95% CI: 3.74, 5.86) [10]. Women in the Dieckmann cohort and from two DESAD sites (Los Angeles and Boston) received high DES doses, and women at the other three DESAD sites (Texas, Minnesota, and Wisconsin) received low DES doses [7,10]. In France, Tournaire et al. found a PTB incidence of 24.22% in the exposed group and of 3.35% in the unexposed group (*p* < 0.001) [13]. More recently, the study by Yim et al. on 54,334 grandmother (G0)-mother (G1) pairs included in the Nurses’ Health Study II showed that the grandmothers’ (G0) treatment with DES during pregnancy was associated with an increased risk of PTB in the G2 generation (adjusted OR (aOR) 2.88; 95% CI [2.46, 3.37]) [41]. However, only our study analyzed PTB and BW in the children of DES sons. Actually, in our study, PTB rate was similar in children of pre-DES, DES, and post-DES sons.

In our cohort, BW was comparable in pre-DES, DES, and post-DES grandchildren (G2, Table 6). Conversely, the overall LBW rate (all births) tended to increase in children of women exposed in utero to DES, but not in children of men exposed in utero to DES. We observed similar results also for post-DES maternal germline children [7]. As in the present study, Yim et al. found that LBW risk was increased in children of women exposed prenatally to DES (aOR 3.09, 95% CI [2.57, 3.72]), but this risk was less important when the analysis was restricted to full-term term births (aOR 1.59; 95% CI [1.08, 2.36]) [41].

In summary, our data are consistent with the notion that DES is one of the first examples of a transplacental toxicant in humans [1,42,43], capable of adversely affecting pregnancy outcomes also in the next generations [1,14,15,16,17,19,36,42,43,44,45,46,47,48]. Fetal DES exposure induces changes in the epigenome that can alter gene expression in a persistent manner, affecting tissue development and function at birth and later in life [49]. Moreover, DES affects the epigenetic reprogramming of the fetal germline (sperms and eggs). In turn, germline epimutations can increase disease susceptibility in the subsequent generations [2].

This study has potential limitations. The first concerns the selection bias, known as the non-responder or non-participation bias. It is not hard to imagine that families with PTB or LBW might have been more interest in participating in this study. This risk was minimized by the high response rate (89.2% in G0 and 96.9% in G1). The second limitation concerns the recall bias that might lead to inaccurate results. To limit this effect in our study, families included in the HHORAGES-France Association were asked to complete two simple questionnaires designed to collect information on major milestones in the women’s history (e.g., delivery date, BW, PTB, major health problems, smoking status, jobs). Moreover, we confirmed data regarding pharmacological treatments, major illnesses, DES exposure, pregnancies, and pregnancy outcomes by medical records. The main limitation of our study is the small sample size. Among the strengths of this study is its design that allowed: (i) ruling out genetic, endocrine, and environmental factors that might have influenced BW; and (ii) excluding all potential changes in EDC exposure in each woman between gestations, in order to better analyze the effect of DES exposure on birth outcomes. Another major strength is that data on total DES dose were supported by the health record or a physician’s note. Moreover, no previous study analyzed PTB and BW in the children of DES sons.

## 4. Subjects and Methods

### 4.1. Study Population

This study was based on a French national retrospective cohort of DES-treated women (HHORAGES-France Association) (n = 529) and their families. The women joined this association for other reasons than PTB, LBW, or HBW. The HHORAGES-France Association contacted them and asked whether they accepted to participate in this study by completing two questionnaires. The first questionnaire was designed to collect information on pregnancy outcomes (sex, pregnancy duration, and BW) and adverse events associated with DES exposure on three generations: the women treated during pregnancy (DES-treated women; generation 0, G0), their in utero-exposed children (DES children; G1), and the children of DES children (DES grandchildren; G2). The first questionnaire included questions on: number of pregnancies/viable offspring/abortions (POA), DES exposure during pregnancy (dose in mg and treatment duration), birth term status (in GW), BW, sex of each viable child, pregnancy complications and outcomes, and subsequent maternal and offspring health problems.

The answers to this on-line questionnaire were used to select women who met the following inclusion criteria: (1) at least three singleton pregnancies with three viable babies (same father) among whom the first child was not exposed in utero to DES (pre-DES), followed by one or more children with fetal exposure to DES (DES) and by one or more children born after the DES treatment (post-DES) (G1); (2) at least one grandchild born from one of her pre-DES or post-DES children and one from her DES child(ren) (G2); and (3) confirmed data on the total DES dose (health report or physician’s observation).

DES-treated women who met these inclusion criteria received a second questionnaire containing questions on their and their progeny’s health history (lifestyle habits, pharmacological treatments, major illnesses or other life events, DES exposure), pregnancies, and pregnancy outcomes. If they did not respond after two consecutive e-mail messages, a trained interviewer called them. If they accepted, the interviewer helped them to fill in the questionnaire during the telephone call (or during another telephone call at their convenience). Data regarding pharmacological treatments, major illnesses, DES exposure, pregnancies, and pregnancy outcomes were confirmed by reviewing the medical records. The local university hospital ethics committee approved this study (IRB-MTP_2022_05_202201113), and all participants gave their informed consent through the HHORAGES-France association.

LBW and HBW were defined as a birth weight ≤2499 g and ≥4000 g, respectively, regardless of the gestational age. PTB was defined as a live birth before 37 weeks of pregnancy.

### 4.2. Covariates

Maternal illnesses (e.g., diabetes, autoimmune diseases), unhealthy lifestyle habits (tobacco, alcohol, drugs), some professions (e.g., hairdressers, cleaners), and some drugs (antiepileptic drugs) can affect fetal growth and are considered potential confounding variables when assessing EDC effect on BW [26]. To rule out all confounding factors, all mothers (G0 and G1) with somatic/psychological disorders, changes in habitat (city vs. countryside), lifestyle habits/job at risk of EDC exposure [50,51], and/or with treatments between pregnancies (e.g., antiepileptic drugs), and also all mothers who reported pregnancy-related problems (i.e., preeclampsia or eclampsia) were excluded. Therefore, only the mother’s age, POA, and in utero DES exposure (yes/no) differed among siblings. The BW of pre-DES, DES and post-DES children from the same parents were analyzed in all G1 and G2 newborns. Twins, stillbirths, and neonatal deaths were excluded for the PTB and BW analyses.

### 4.3. Statistical Analysis

Descriptive statistics are presented as means (SD) or numbers (%), as appropriate. Continuous variables were transformed into categorical variables using clinical thresholds.

The Cochran-Armitage test was used to assess whether there was a linear trend for PTB, LBW and HBW in the three groups (pre-DES, DES, and post-DES). A multivariate mixed model was used to select determinants of the three defined outcomes: PTB, LBW, and HBW. The outcome was adjusted for clustering within mothers. Data on BW were compared with the two-tailed Wilcoxon rank sum test. P values <0.05 were considered significant and all statistical tests were two-sided. Statistical analyses were conducted with SAS V.9.4 (SAS Institute. Cary, NC, USA).

## 5. Conclusions

Although our cohort is small, our results show the effects of DES contamination in the same informative families and suggest a probable persisting contamination in post-DES pregnancies (children and grandchildren). The study of DES effects has contributed to new pathophysiological hypotheses concerning the fetal environment role in the development of chronic diseases in adults (reproductive problems, metabolism disorders, hormone-dependent cancers, neuropsychiatric disorders) [52]. Therefore, clinicians must continue to identify DES children and grandchildren and offer them appropriate management and follow-up.

This retrospective observational study strengthens the suspicion of the multigenerational effects of EDCs and reinforces the need of more studies on the long-term outcomes of in utero exposure to EDCs.

## Figures and Tables

**Figure 1 ijerph-20-02542-f001:**
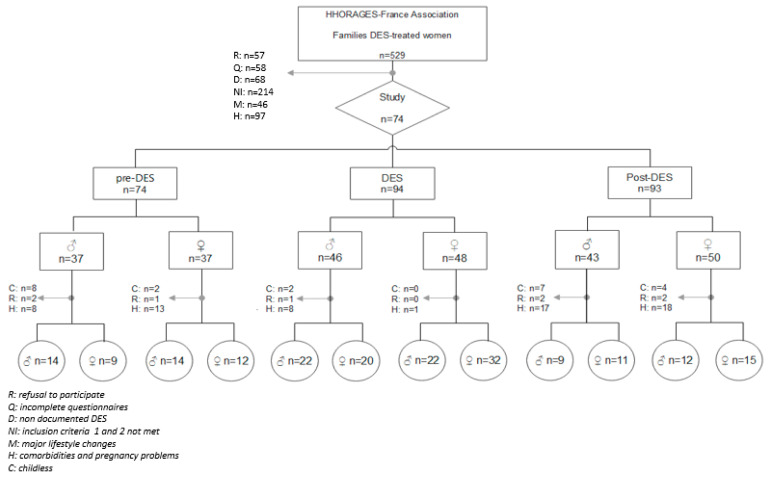
Flowchart of the families included in the study. More than one answer was possible.

**Table 1 ijerph-20-02542-t001:** Characteristics of the excluded families.

Exclusion Criteria	n
Declined to participate	57
Incomplete questionnaire	54
Only one delivery	49
Only two deliveries	60
Three or more deliveries, but not pre-DES, DES, and post-DES conditions	47
Three or more deliveries with pre-DES, DES, and post-DES conditions, but at least one condition represented only by twins, stillbirth, or neonatal death	14
Three or more deliveries with pre-DES, DES, and post-DES conditions, but without at least one grandchild born from one of her pre-DES or post-DES children and one from her DES child	33
Changed partners	11
Not documented DES exposure	68
Comorbidities	
Diabetes mellitus	5
Thyroid disease	7
BMI <18.5 or >25	9
Severe anemia	1
Chronic hypertension	1
Chronic renal disease	1
Chronic cardiorespiratory disease	1
Autoimmune disease	7
Epilepsy with anti-epileptic drugs	2
Cancer	3
History of caesarean section	15
Severe depression	5
Changed habitat between pregnancies (city, countryside)	12
Changed lifestyle habits between pregnancies (smoke, alcohol, illicit drugs)	14
Changed job with EDC exposure between pregnancies (beautician, cleaner, hairdresser, laboratory technician)	17
Changed anti-epileptic drugs between pregnancies	3
Presented pregnancy-related problems	
Gestational diabetes mellitus	5
Intrahepatic cholestasis	2
Gestational hypertension	8
Preeclampsia	12
Eclampsia	1
Infections	0
Traumatisms	0
Major congenital anomalies in fetus	3
Major life event during pregnancy (divorce, a death in the family, injury, or job loss)	9

Legend: Changes in maternal professions that are at risk of potential exposure to EDCs according to the Job-exposure Matrix by Tongeren et al. [13]: cleaners, domestics, jobs using solvents, chemists, laboratory technicians, photographers and audiovisual equipment operators, painters and decorators, jobs in cosmetics, hairdressers, barbers, beauticians and related occupations, agricultural jobs using pesticides, farm managers, agricultural and fishing trades, officers in armed forces, police officers (sergeant and below), electrical/electronic technicians, electrical/electronic engineers, textiles, garments, dental nurses, waiters, waitresses, war park attendants.

**Table 2 ijerph-20-02542-t002:** Characteristics of G0 mothers and G1 children in function of their DES exposure status.

	Pre-DES					DES					Post-DES				
Variable	n	Mean ± SD	Median	Min	Max	n	Mean ± SD	Median	Min	Max	n	Mean ± SD	Median	Min	Max
Mean DES dose	-	-	-	-	-	94	3982.4 ± 412.4	4200	2100	4250	-	-	-	-	-
IPI	-	-	-	-	-	94	1.4 ± 0.4	1.4	0.4	2.3	93	1.4 ± 0.5	1.4	0.6	2.4
Maternal age	74	23.6 ± 2.7	23.55	19.4	31.1	94	26.3 ± 3.1	25.9	21.2	34.2	93	29.1 ± 3.3	28.7	23.2	38.2
GW	74	39.8 ± 1.1	40	33	40	94	38.5 ± 3.1	40	28	40	93	39.3 ± 2.1	40	30	40
BW	74	3203.4 ± 423.5	3100	1850	4200	94	3141.1 ± 854.1	3335	980	4900	93	3347.4 ± 634	3345	1260	4200
BW ≥37 GW	72	3240.3 ± 365.1	3100	2650	4200	80	3421.0 ± 543.9	3500	2200	4900	83	3485.3 ± 493.0	3401	1700	4200

Legend: DES = diethylstilbestrol; IPI = interpregnancy interval; GW = gestational week; BW = birth weight; BW ≥ 37 GW = birth weight of full-term newborns.

**Table 3 ijerph-20-02542-t003:** Characteristics of G1 boys and girls in function of their DES exposure status.

	Pre-DES		DES		Post-DES		*p* Value Cochran-Armitage Test for Trend
	BOYS (n = 37)	GIRLS (n = 37)	BOYS (n = 46)	GIRLS (n = 48)	BOYS (n = 43)	GIRLS (n = 50)	
PTB (< 37 GW), n (%)	2 (2.7)		14 (14.9)		10 (10.8)		0.0095
LBW < 2499 g, n (%)	0		2 (2.9)		2 (2.5)		0.0006
HBW > 4000 g, n (%)	3 (4.17)		8 (11.6)		18 (22.5)		0.0033

Legend: DES = diethylstilbestrol; GW = gestational week; PTB = pre-term birth; LBW = low birth weight; HBW = high birth weight.

**Table 4 ijerph-20-02542-t004:** Multivariate modeling of PTB, LBW, and HBW in G1 children.

	PTB	LBW < 2499 g	HBW > 4000 g
Variables	*p*-Value	*p*-Value	*p*-Value
Current smoker	0.7362	0.3932	0.3845
Sex	0.8542	0.3608	0.7394
GW		<0.0001	0.0616
Mother age	0.5596	0.7272	0.5772
IPI	0.7800	0.6678	0.0948
Exposure group	0.4480	0.0269	0.0005
Prescribed DES dose	0.1304	0.0661	0.3456
Number of pregnancies	0.1395	0.0640	0.0087

Legend: DES = diethylstilbestrol; GW = gestational week; PTB = pre-term birth; LBW = low birth weight; HBW = high birth weight; IPI = interpregnancy interval; Exposure group = DES group vs. pre-DES and post-DES groups.

**Table 5 ijerph-20-02542-t005:** Characteristic of G1 mothers (G1 daughters or partners of G1 sons, when adults) excluded from the analysis. More than one answer was possible.

		Pre-DES		DES		Post-DES	
		BOYS	GIRLS	BOYS	GIRLS	BOYS	GIRLS
Comorbidities	BMI < 18.5 or > 25	2	1	3		2	3
	Severe anemia			1			
	Chronic hypertension				1		1
	Chronic cardiorespiratory disease					1	
	Chronic renal disease						1
	Autoimmune disease	1				2	2
	Diabetes mellitus		1				
	Thyroid problem		2			1	2
	Epilepsy with anti-epileptic drugs		1			1	
	Cancer		1				
	Severe depression		2			4	3
Pregnancy-related problems	Gestational diabetes mellitus	2	3	3	1	1	2
	Gestational hypertension	2	2			2	3
	Preeclampsia	1		2		2	1
	Intrahepatic cholestasis		1				
	Major congenital anomalies in fetus						

Legend: DES = diethylstilbestrol; BMI = body mass index.

**Table 6 ijerph-20-02542-t006:** Characteristics of G2 boys and girls in function of their DES exposure status.

	Pre-DES	DES	Post-DES
	Paternal Germline	Maternal Germline	Paternal Germline	Maternal Germline	Paternal Germline	Maternal Germline
Mothers (n)	19	21	35	47	17	26
Maternal age	24.5 ± 2.3	25.6 ± 3.0	24.6 ± 2.4	25.8 ± 2.9	26.7 ± 2.8	25.8 ± 3.0
G2 children (n)	23	26	42	54	20	27
Sex G2 children (n)	14 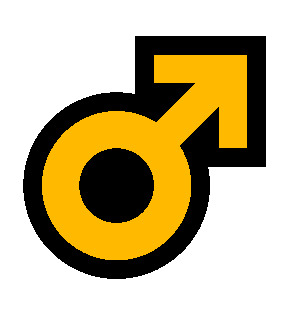	9 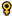	14 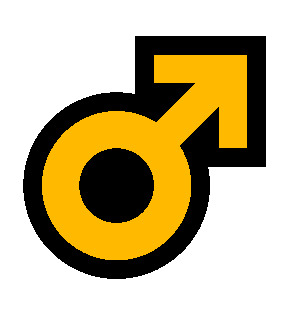	12 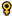	22 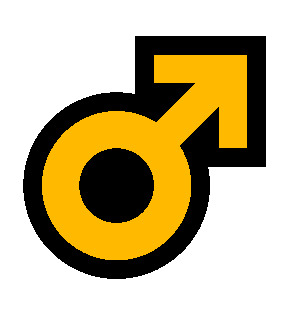	20 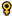	22 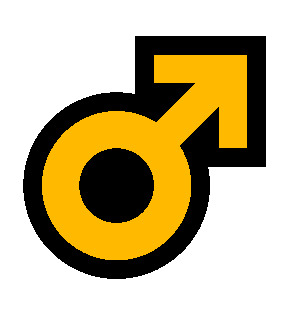	32 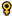	9 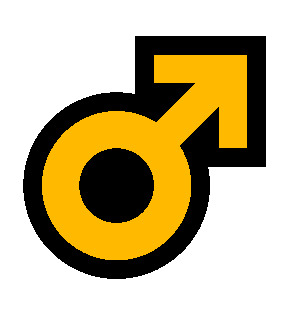	11 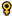	12 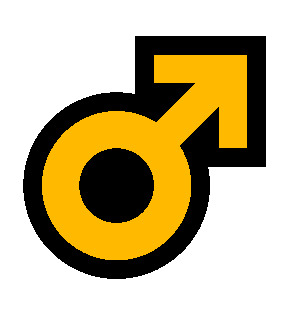	15 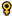
PTB (GW < 37 weeks) n (%)	2	(8.7)	2	(7.7)	0	(0)	8	(14.8)	2	(10.0)	4	(14.8)
LBW n (%)	2	(8.7)	2	(7.7)	0	(0)	8	(14.8)	2	(10.0)	3	(11.1)
LBW in ≥ 37 GW n (%)	0	(0)	2	(8.3)	0	(0)	0	(0)	0	(0)	0	(0)
HBW n (%)	4	(17.4)	2	(7.7)	2	(4.8)	6	(11.1)	0	(0)	0	(0)
HBW in ≥ 37 GW n (%)	4	(19.1)	2	(8.3)	2	(4.8)	6	(13.0)	0	(0)	0	(0)
BW (n), mean *±* SD (g)	23	3313.0 ± 611.9	26	3218.5 ± 415.6	42	3329.2 ± 418.6	54	3214.3 ± 693.5	20	3228.5 ± 473.3	27	3185.4 ± 516.5
BW ≥ 37GW (n), mean *±* SD (g)	21	3443.8 ± 450.4	24	3245.8 ± 421.4	42	3329.2 ± 418.6	46	3422.9 ± 500.0	18	3353.3 ± 289.0	23	3344.6 ± 332.6

Legend: DES = diethylstilbestrol; PTB = pre-term birth; GW = gestational week; LBW = low birth weight; HBW = high birth weight; BW = birth weight.

## Data Availability

Data regarding any of the subjects in the study has not been previously published.

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
