# Peer review of "Birth Outcomes in DES Children and Grandchildren: A Multigenerational National Cohort Study on Informative Families"

_ijerph, 2023, doi:10.3390/ijerph20032542_

Round 1
Reviewer 1 Report
Manuscript is well written, study is well modeled and results are well presented. Only one correction should be made.
Introduction
Line 54-multigenerational should be replaced with transgeneraitonal as this experiment does not show effect in several generations
Author Response
REVIEWER 1:
Q : Line 54-multigenerational should be replaced with transgenerational as this experiment does not show effect in several generations
According to Nilsson et al., the definition of epigenetic transgenerational inheritance is “germline-mediated inheritance of epigenetic information between generations in the absence of continued direct environmental influences that leads to phenotypic variation”. Conversely, multigenerational exposure refers to the effects observed in subsequent generations that are the result of direct exposure. Direct environmental exposure of the parents, considered to be the F0 generation, can also affect their germline (sperm or eggs). The next generation (F1), derived from this germline, is still considered exposed, and therefore it is not truly transgenerational. For preconception parental exposure, the F2 generation offspring is considered the first transgenerational unexposed generation. The situation is different when a pregnant woman is exposed, because the fetus and its germline also are directly exposed. In that case, the F3 generation is the first unexposed transgenerational offspring. For this reason, we used the term “multigenerational” and not “transgenerational”.
Nilsson EE, Sadler-Riggleman I, Skinner MK. Environmentally induced epigenetic transgenerational inheritance of disease. Environ Epigenet. 2018 Jul 17;4(2):dvy016. doi: 10.1093/eep/dvy016. PMID: 30038800; PMCID: PMC6051467.
Reviewer 2 Report
1. Is there any difference between "1.4±0.4" in line 163 and "1.4±0.5" in line 165;
2. Is the expression "but comparable between DES and post-DES children (p = 0.3645)" in rows 184-185 and 263 correct;
3. The bandwidth in Table 2 > 37GW and the last row in Table 6 indicate an error;
4. Whether the BW of line 235 should be changed to LBW;
5. The keyword part can be considered again;
6. Whether there are differences in DES exposure during different pregnancy;
7. The mechanism of DEC affecting male and female offspring;
8. The mechanism of DEC and the effects of its metabolites on the body;
9. Mechanism of DEC-induced preterm birth and BW alteration;
10. Whether there is a genetic change in the cumulative effect of DEC on offspring should be considered;
11. The results from rows 33 to 35 should be further analyzed: "Preterm birth (PTB) was 2.7% before DES, 14.9% after DES, 33% and 10.8% after DES (Cochran-Armitage test for trend, p=0.0095). Mean BW was 34 (≥37 weeks gestation) higher in DES than in pre-DES term neonates (p=0.007)";
12. The reasons why our results are inconsistent with previous studies should be further analyzed;
13. The results expressed in lines 228 to 230 should be further analyzed in the Discussion;
14. Give corresponding diagnosis and treatment suggestions to patients after DEC exposure.
Author Response
REVIEWER 2:
Q1. Is there any difference between "1.4±0.4" in line 163 and "1.4±0.5" in line 165;
No, we thank the reviewer for highlighting this mistake. We corrected the manuscript.
Q2: Is the expression "but comparable between DES and post-DES children (p = 0.3645)" in rows 184-185 and 263 correct;
Following the reviewer’s suggestion, we modified the manuscript as follows: “BW was similar in DES and post-DES children”.
Q3: The bandwidth in Table 2 > 37GW and the last row in Table 6 indicate an error;
Please, note that the data included in Tables 2 and 6 are related to children and grand-children, respectively.
Q4: Whether the BW of line 235 should be changed to LBW;
One of the main message of our study is that the mean BW was higher in DES than in pre-DES full-term neonates, so we think that “BW” should be maintained in line 235.
Q5: The keyword part can be considered again;
We do not see what is wrong with the key words.
Q6: Whether there are differences in DES exposure during different pregnancy;
No, the 20 women who had two DES children used the same schedule and dose for the two pregnancies. We thank the reviewer for this remark. We added the following sentence in the manuscript: “When mothers had more than one DES pregnancy, the schedule and prescribed DES doses were the same for all pregnancies”.
Q7: The mechanism of DES affecting male and female offspring;
Following the reviewer’s comment, we added these sentences in the manuscript: “Fetal DES exposure induces changes in the epigenome that can alter gene expression in a persistent manner, affecting tissue development and function at birth and later in life (50). Moreover, DES affects the epigenetic reprogramming of the fetal germline (sperms and eggs). In turn, germline epimutations can increase disease susceptibility in the subsequent generations (2).”
Q8: The mechanism of DES and the effects of its metabolites on the body;
Troisi et al. reported that prenatal DES exposure was associated with a modestly increased risk of diabetes in women (HR 1.5, 95% CI 1.0–2.1) but not in men. This effect is an adult consequence of fetal DES exposure and we did not include this topic in our manuscript.
Q9: Mechanism of DES-induced preterm birth and BW alteration;
Following the reviewer’s comment, we added some sentences in the manuscript: “The possible mechanisms underlying the higher risk of PTB and of LBW reported in most studies on DES children remain speculative. According to Hatch et al., DES may interfere with the delicate balance of pregnancy hormones, such as the ratio of estradiol to estriol (two estrogens), and this might alter the timing of parturition (7, 31). DES may also induce maternal and fetal stress, leading to an increase in the secretion of corticotropin-releasing hormone, associated with preterm labor and premature rupture of membranes (7, 32-35).”
Q10: Whether there is a genetic change in the cumulative effect of DES on offspring should be considered;
Following the reviewer’s comment, we added these sentences in the manuscript: “Fetal DES exposure induces changes in the epigenome that can alter gene expression in a persistent manner, affecting tissue development and function at birth and later in life (50). Moreover, DES affects the epigenetic reprogramming of the fetal germline (sperms and eggs). In turn, germline epimutations can increase disease susceptibility in the subsequent generations (2).”
Q11: The results from rows 33 to 35 should be further analyzed: "Preterm birth (PTB) was 2.7% before DES, 14.9% after DES, 33% and 10.8% after DES (Cochran-Armitage test for trend, p=0.0095). Mean BW was 34 (≥37 weeks gestation) higher in DES than in pre-DES term neonates (p=0.007)";
We agree with the reviewer’s comment. Unluckily, we have already used the total number of words for the Abstract. Conversely, we extensively discussed these results in Discussion section.
Q12: The reasons why our results are inconsistent with previous studies should be further analyzed;
The possible mechanisms underlying the higher BW found in full-term newborns in the DES than pre-DES group, unlike previous data from American cohorts, remain speculative. We modified the paragraph as follows: “In addition, the higher BW found in our cohort of DES children compared with previous literature data was predictable. Indeed, the outcomes observed in women treated with DES during pregnancy can be accurately replicated in the mouse (36). For instance, in pregnant mice exposed to a 4-log range of DES concentrations from day 11 to day 17 of gestation, Palanza et al. did not observe any PTB, but delayed parturition in 50% and 100% of mice treated with 10 µg and 100 µg DES, respectively (37). DES did not have any significant effect on BW; however, the dose-response relationship was complex (non-monotonic biphasic), and opposite effects (i.e. higher or lower BW) were observed as DES doses increased (37). In women, the total DES doses prescribed ranged between <100 mg and up to 46,600 mg (38). In the USA, “high dose cohorts” with median DES doses of 12,442 (Chicago), 8,575 (Boston), and 7,550 mg (California) have been monitored, as well as “low dose cohorts” with median doses of 3,175 (Wisconsin), 2,572 (Texas), and 1,520 mg (Minnesota) (39, 40). The median total dose in our HHORAGES-France cohort and in another French cohort described by Tournaire et al. was ~4,000 mg (40). Therefore, in our cohort, fetuses were exposed to DES doses between those of the American “low-dose” and “high-dose cohorts”. We may hypothesize that BW variations may follow a similar non-monotonic biphasic relationship as observed in the mouse model. This might explain why in our cohort, BW in full-term newborns was higher in the DES than pre-DES group, unlike previous findings in American cohorts (7, 24).”
Q13: The results expressed in lines 228 to 230 should be further analyzed in the Discussion;
Following the reviewer’s comment, we added the following sentence in the manuscript: “Actually, in our study, PTB rate was similar in children of pre-DES, DES, and post-DES sons.”
Q14: Give corresponding diagnosis and treatment suggestions to patients after DES exposure.
We agree with the reviewer’s comment and added some sentences in the manuscript: “The study of DES effects has contributed to new pathophysiological hypotheses concerning the fetal environment role in the development of chronic diseases in adults (reproductive problems, metabolism disorders, hormone-dependent cancers, neuropsychiatric disorders) (51). Therefore, clinicians must continue to identify DES children and grandchildren and offer them appropriate management and follow-up.”
Reviewer 3 Report
Dear Editor,
thank you for giving me the opportunity to review this paper on the long-term effects of administrations of DES in pregnant women. Despite the topic is not really innovative, I believe that the authors made a great effort in collecting all the available data on three generations of women exposed to DES. The overall number of included cases is quite low and far from being appropriate for definitive conclusions, however the data may be useful in managing pregnancy of third generation women exposed to DES.
I don't have any particular comment to make and I believe that the manuscript is suitable for publication.
Author Response
OK
Reviewer 4 Report
The manuscript “Birth outcomes in DES children and grandchildren: A multi-2 generational national cohort study on informative families” describes pregnancy duration and birthweight variations in the children and grandchildren born before, during, and 23 after maternal DES treatment in the same informative families, to rule out genetic, endocrine and 24 environmental factors.
A very interesting piece of work and very well done. Despite the small number of cases, the influence of the drug on subsequent generations could be shown very clear. the processing of the data is very clean and the results are clearly described.
The tables are informative and clear.
The text is written clearly and understandably.
the only thing that would be desirable is a more detailed description of the effects of DES when taken during pregnancy and the effect on the unborn child in the Introduction section
Author Response
REVIEWER 4:
Q1 : the only thing that would be desirable is a more detailed description of the effects of DES when taken during pregnancy and the effect on the unborn child in the Introduction section.
We agree with the reviewer’s comment and added some sentences in the manuscript: “It is well-known that in utero exposure to DES induces a wide range of reproductive tract abnormalities in ‘DES daughters’, such as alterations of Müllerian duct development, fertility problems, ectopic pregnancies, miscarriages, premature births, endometriosis and cancers, particularly clear cell adenocarcinoma of the vagina and cervix in girls and young women (5-15). In ‘DES sons’, epididymal cysts, hypospadias, cryptorchidism, hypoplastic testis and micro-penis have been reported (16, 17). Besides such somatic effects, prenatal DES exposure has been associated also with psychiatric disorders, such as severe depression, behavioral disorders, eating disorders, schizophrenia, bipolar disorders, anxiety, suicide and suicide attempts, as well as neurodevelopmental disorders, including autism spectrum disorder (18-21).”